# Cost-effectiveness and benefits of perinatal health interventions in high-income settings: A protocol for a systematic review of economic evaluations

Tsegaye G. Haile [1,2] *, Gizachew A. Tessema [1,3,4], Lucas Hertzog[1], Elizabeth Newnham[1,3], Berihun Assefa Dachew[1,3], Marshall Makate[1]

1 Curtin School of Population Health, Curtin University, Perth, WA, Australia, 2 Department of Health Systems and Policy, Institute of Public Health, University of Gondar, Gondar, Ethiopia, 3 enAble Institute, Curtin University, Perth, WA, Australia, 4 School of Public Health, University of Adelaide, Adelaide, South Australia, Australia

* t.haile2@postgrad.curtin.edu.au

**Data Availability Statement:** No datasets were generated or analysed during the current study. All

## Abstract

### Background

Despite ongoing efforts, perinatal morbidity and mortality persist across all settings, imposing a dual burden of clinical and economic strain. Besides, the fragmented nature of economic evidence on perinatal health interventions hinders the formulation of effective health policies. Our review aims to comprehensively and critically assess the economic evidence for such interventions in high-income countries, where the balance of health outcomes and fiscal prudence is paramount.

### Methods and analysis

We will conduct a comprehensive search for studies using databases including EconLit (EBSCO), Cost Effectiveness Analysis (CEA) Registry, Medline (Ovid), Embase (Ovid), CINAHL Ultimate (EBSCO), Global Health (Ovid), and PubMed. Furthermore, we will broaden our search to include Google Scholar and conduct snowballing from the final articles included. The search terms will encompass economic evaluation, perinatal health interventions, morbidity and mortality, and high-income countries. We will include full economic evaluations focusing on cost-effectiveness, cost-benefit, cost-utility, and cost-minimisation analyses. We will exclude partial economic evaluations, reports, qualitative studies, conference papers, editorials, and systematic reviews. Date restrictions will limit the review to studies published after 2010 and those in English during the study selection process. We will use the modified Drummond checklist to evaluate the quality of each included study. Our findings will adhere to the Preferred Reporting Items for Systematic Reviews and Meta-analyses (PRISMA) 2020 statement. A summary will include estimated costs, effectiveness, benefits, and the incremental cost-effectiveness ratio (ICER). We also plan to conduct a subgroup analysis. To aid comparability, we will standardise all costs to the United States

relevant data from this study will be made available upon study completion.

**Funding:** The author(s) received no specific funding for this work.

**Competing interests:** The authors have declared that no competing interests exist.

Dollar, adjusting them to their 2022 value using country-specific consumer price index and purchasing power parity.

## Ethics and dissemination

This systematic review will not involve human participants and requires no ethical approval. We will publish the results in a peer-reviewed journal.

## Trial registration

We registered our record on PROSPERO (registration #: CRD42023432232).

## Background

Adverse perinatal outcomes include morbidity and mortality occurring from viability to 28 days after birth [1]. Perinatal morbidity encompasses preterm births (births occurring before 37 weeks of gestational age), low birth weight (LBW) (birth weight less than 2500 grams), small-for-gestational-age (SGA) (birth weight below the 10th percentile), and congenital anomalies. Perinatal mortality includes stillbirths (foetal death with a weight higher than 400 grams) and neonatal deaths within the first four weeks of life [1].

Despite global efforts to address adverse perinatal outcomes, reducing perinatal morbidity and mortality remains a significant public health challenge, with an estimated 2 million stillbirths and 2.4 million neonatal deaths occurring annually [2]. In this context, the United Nations' Sustainable Development Goals (SDGs) target 3.2 aims to end all preventable deaths, with specific targets to reduce neonatal deaths to 12 per 1,000 live births or fewer and under-five deaths to fewer than 25 per 1,000 live births by 2030 [3,4]. The Every Newborn Action Plan (ENAP) aims to decrease the global stillbirth rate to fewer than ten stillbirths per 1,000 births by 2035 [5].

Global disparities in perinatal mortality rates range from the lowest in northern Europe to the highest in sub-Saharan Africa [6]. Significant disparities between population groups exist even in countries with generally low perinatal mortality rates. While adverse perinatal outcomes are disproportionately higher in resource-limited settings, it is still a significant burden in high-income countries, and disparities exist across different socioeconomic classes, residential areas, and racial groups within the country. For example, in Australia in 2021, despite the national prevalence of perinatal mortality being 9.6 deaths per 1,000 live births, the number climbs to 19 deaths per 1,000 births among those births from women in very remote areas and 17 deaths per 1,000 births among those from women of Aboriginal and Torres Strait Islander background [7]. Perinatal morbidity and mortality impose consequential health and economic burdens on individuals, families, and society [8,9]. It also impacts the healthcare system in a competitive, resource-limited world. In response, the healthcare system has implemented various health interventions. These include but are not limited to, the provision of aspirin for the prevention of pre-eclampsia [10], increased ultrasound surveillance to detect growth restriction [11] and ensure timely delivery to prevent preterm birth and stillbirth, progesterone or cervical cerclage to prevent preterm birth [12,13], earlier screening and treatment for gestational diabetes, induction of labour near term, antenatal steroids to reduce respiratory distress syndrome and death, and transfer to higher level care for neonatal intensive care unit (NICU) [14].

These interventions reduce perinatal mortality and morbidity, showing promising survival and outcome benefits through universal health coverage [15,16]. Reports consistently highlight a considerable burden on the national health system due to the high utilisation of healthcare resources caused by adverse perinatal outcomes [17–19]. However, the investigation into resource utilisation for health interventions designed to avert perinatal morbidity and mortality remains notably scant.

Despite the perceived benefits of health interventions in reducing perinatal morbidity and mortality, no study has yet offered a comprehensive and systematic synthesis of the available economic evaluation evidence. Evaluating perinatal health interventions from both health and economic perspectives will be warranted in identifying the most effective healthcare interventions/strategies to avert mortality and morbidity and maximise health benefits. The study will assist health decision-makers in understanding the benefits, costs, and cost-effectiveness of portfolios of interventions designed to reduce perinatal mortality and mortality. Whenever possible, the study will also evaluate perinatal health interventions across the continuum of care from preconception through the early childhood period.

While high-income countries have substantially improved perinatal health, disparities among various groups remain. Addressing these inequalities requires the tailoring of healthcare interventions to meet specific challenges. However, implementing cost-effective interventions that reduce these disparities has yet to receive sufficient attention within the healthcare system. Recognising this oversight early on is vital for future cost savings in the health sector. Moreover, differences in healthcare infrastructure, resources, access to care, and policy priorities contribute significantly to disparities in perinatal health outcomes between high-income and other settings. Therefore, it is crucial to examine evidence on the cost-effectiveness and benefits of perinatal health interventions separately.

In today's context, with perinatal morbidity and mortality remaining significant in low-middle-income countries, most of the perinatal health intervention studies focus on these regions, resulting in a lack of comprehensive evidence on the economic evaluations of perinatal health in high-income nations. Therefore, we aim to systematically summarise findings from comprehensive evidence synthesis on the cost-effectiveness and benefits of perinatal health interventions in high-income countries. We aim to evaluate the most effective interventions and provide the highest value in reducing perinatal morbidity and mortality. Given the dynamic nature of the healthcare landscape, with constant changes in interventions, strategies, regulations, and patient characteristics, recent cost-effectiveness and cost-benefit data are essential for reflecting the current healthcare environment. Given that the costs of interventions and their outcomes can vary significantly over time due to changes in the health system, time horizon, inflation, and market conditions, we consider data prior to 2010 increasingly unreliable for understanding current patterns. Additionally, data from before 2010 may pose comparability challenges due to the extensive period involved. Therefore, we will focus our review on studies from 2010 onwards.

## Methods

### Protocol design

We intend to systematically review published economic evaluation studies about the cost-effectiveness, cost-benefit, and cost-utility analysis of perinatal health interventions. Our review will adhere to the Preferred Reporting Items for Systematic Reviews and Meta-Analyses (PRISMA) 2020 statement [20]. Furthermore, we will document any significant modifications to this protocol and plan to publish them alongside the review results. We have registered the

review protocol in the Prospective International Register of Systematic Reviews (PROSPERO) database (registration number: CRD42023432232).

## Search strategy

We will conduct comprehensive searches across major databases to identify relevant studies, including EconLit (EBSCO), the Effectiveness Analysis (CEA) Registry, Medline (Ovid), Embase (Ovid), CINAHL Ultimate (EBSCO), Global Health (Ovid), and PubMed. Additionally, Google Scholar will be used to explore further literature. Moreover, any additional pertinent studies referenced within the included articles will be evaluated and incorporated into the final review through citation snowballing.

We will use subject headings and keywords to formulate search terms corresponding to broader categories:

i. Economic evaluation, incorporating terms such as cost-effectiveness, cost-benefit, cost-minimisation, and costs of interventions.

ii. Perinatal health interventions encompassing perinatal care, neonatal intensive care, and prophylaxis during pregnancy to prevent adverse neonatal outcomes.

iii. Perinatal morbidity and mortality, addressing preterm births, SGA, LBW, congenital anomalies, stillbirths, and neonatal deaths.

iv. High-income countries (Table 1 for a sample MEDLINE search strategy).

We will restrict our study selection to English-language literature published in peer-reviewed journals. We will focus on articles reporting the cost-effectiveness, cost-benefit, and cost of perinatal health interventions aimed to reduce adverse outcomes. Furthermore, we will include studies that report on full economic evaluations, which encompass cost-effectiveness, cost-utility, cost-benefit, and cost-minimisation analyses. These evaluations should primarily address direct or potential interventions spanning from pregnancy through the early childhood period up to the age of five. We will exclude partial economic evaluations, reports, qualitative studies, conference papers, editorials, and systematic reviews from our review. We will apply date restrictions during the study selection process, limiting the review to studies published after 2010.

## Study selection

All retrieved articles will initially be exported to EndNote 20 to compile all records identified during the search and facilitate the removal of duplicates. Subsequently, we will export records to Rayyan, a systematic review management software package, for title and abstract screening [21]. The review process will follow a two-step approach, wherein paper selection will adhere to predetermined eligibility criteria. Initially, two reviewers will independently screen titles and abstracts against the selection criteria. Subsequently, the full text of relevant records will undergo independent assessment based on the eligibility criteria. Any discrepancies during the screening process will be resolved through discussion between reviewers. Finally, we will present the study selection process using the PRISMA flow diagram [20].

## Study eligibility criteria

We use the **P**opulation-**I**ntervention-**C**omparison-**O**utcomes-**T**ype of study-**C**ontext (PICOTC) approach to formulate and address the review questions [22] and delineate the study eligibility criteria.

**Populations.** The population under consideration will encompass studies involving pregnant women beyond 20 weeks of gestation, newborns, infants, and children under five years of age. This review will span from pregnancy to early childhood to evaluate the effectiveness and benefits of perinatal health interventions in mitigating morbidity and mortality associated with adverse perinatal outcomes up to the age of five.

**Interventions.** We will incorporate studies with economic evaluations of perinatal health interventions, including preventive, curative, and health promotional activities to prevent adverse perinatal outcomes. These adverse perinatal outcomes comprise stillbirths, neonatal mortality, preterm birth, LBW, SGA, congenital anomalies, and associated comorbidities.

**Table 1. Medline searching strategy.**

| Terms | CONCEPT 1 | CONCEPT 2 | CONCEPT 3 | CONCEPT 4 |
|---|---|---|---|---|
| | Economic Evaluation | Perinatal health interventions | Adverse perinatal outcomes | High-income countries |
| Subject headings | exp cost-benefit analysis/ or exp cost-effectiveness analysis/ or exp Quality-Adjusted Life Years/ or exp "Quality of Life"/ or exp Cost-Benefit Analysis/ or exp Health Care Costs/ or exp Economics, Pharmaceutical/ or exp Economics, Medical/ or Economics/ or exp Economics, Hospital/ | exp Perinatal Care/ or exp Maternal Health Services/ or exp Prenatal Care/ or exp Delivery, Obstetric/ or exp Infant Care/ or exp Intensive Care Units, Pediatric/ or exp Child Health Services/ | exp Pregnancy Complications/ or exp Pregnancy Outcome/ or exp Fetal Death/ or exp Fetal Diseases/ or exp Perinatal Mortality/ or exp Perinatal Death/ or exp Premature Birth/ or exp Infant, Low Birth Weight/ or exp Congenital Abnormalities/ or exp Infant, Newborn, Diseases/ | exp Developed Countries/ or exp "Organisation for Economic Co-Operation and Development"/ or exp "Scandinavian and Nordic Countries"/ or exp Andorra/ or exp "Antigua and Barbuda"/ or exp Aruba/ or exp Australia/ or exp Austria/ or exp Bahamas/ or exp Bahrain/ or B exp Barbados/ or exp Belgium/ or exp Bermuda/ or exp British Virgin Islands/ or exp Canada/ or exp West Indies/ or exp Channel Islands/ or exp Chile/ or exp Croatia/ or exp Curacao/ or exp Cyprus/ or exp Czech Republic/ or exp Denmark/ or exp Estonia/ or exp Finland/ or exp France/ or exp Polynesia/ or exp Germany/ or exp Gibraltar/ or exp Greece/ or exp Greenland/ or exp Guam/ or exp Hong Kong/ or exp Hungary/ or exp Iceland/ or exp Ireland/ or exp Israel/ or exp Italy/ or exp Japan/ or exp "Republic of Korea"/ or exp Kuwait/ or exp Latvia/ or exp Liechtenstein/ or exp Lithuania/ or exp Luxembourg/ or exp Malta/ or exp Monaco/ or exp Netherlands/ or exp New Caledonia/ or exp New Zealand/ or exp Norway/ or exp Oman/ or exp Panama/ or exp Poland/ or exp Portugal/ or exp Puerto Rico/ or exp Qatar/ or exp Romania/ or exp San Marino/ or exp Saudi Arabia/ or exp Seychelles/ or exp Singapore/ or exp Sint Maarten/ or exp Slovakia/ or exp Slovenia/ or exp Spain/ or exp Sweden/ or exp Switzerland/ or exp Taiwan/ or exp "Trinidad and Tobago"/ or exp "Saint Kitts and Nevis"/ or exp United Arab Emirates/ or exp United Kingdom/ or exp United States/ or exp Uruguay/ or exp United States Virgin Islands/ |

**Table 1.** (Continued)

| Terms | CONCEPT 1 | CONCEPT 2 | CONCEPT 3 | CONCEPT 4 |
|---|---|---|---|---|
| | **Economic Evaluation** | **Perinatal health interventions** | **Adverse perinatal outcomes** | **High-income countries** |
| | (Cost$ ADJ2 (effective$ or utilit$ or benefit$ or consequence$ or minimi$)).tw,ab. Or economic evaluation$.tw,ab. Or (decision ADJ2 (analy$ or model$ or tree$)).tw,ab. or (cost$ or economic$ Or pharmacoeconomic$).ti. Or quality-adjusted life year$.ti,ab,kw. or "Value for Money" OR "Economic evaluation*" OR "Cost Effective Analys$s" OR "Cost Utility Analys$" OR "Cost Benefit Analys$s" OR "Cost Consequence* OR Analys$s" OR "cost minimi$ation analys$s" OR "Return o* Investment" OR "return to investment" OR "Social Return o* Investment" OR "social return to investment" OR "cost effective*" OR efficien* OR "cost saving*" OR "return on" or ((benefits or economic* or financial) ADJ3 (illness* or burden*)).tw,ab. | ((Perinatal or newborn or neonatal or prenatal or obstetric or infant or p$diatric or child or essential or intensive) adj3 (care or service* or health service* or intervention* or promotion* or prevention*)).tw,ab | ((Adverse or perinatal or pregnancy or gestation or birth or neonatal or obstetric or infant or fetal or congenital or child) ADJ3(outcome* or death or defect or anomal* or complication or event or loss or mortalit* or morbidit* or consequence or sequelae)).tw,ab. OR (Stillbirth or still-birth or prematurity or preterm or small for gestational age or small baby or PTB or SGA or low birthweight or LBW).tw,ab. | ((Developed or high income or high-income* or high resource or industrialised or industrialized or nordic or scandinavian) ADJ3 (countr* or nation* or setting*)).tw,ab. OR (Andorra or "Antigua and Barbuda" or Aruba or Australia or Austria or "Bahamas, The" or Bahrain or Barbados or Belgium or Bermuda or "British Virgin Islands" or "Brunei Darussalam" or Canada or "Cayman Islands" or "Channel Islands" or Chile or Croatia or Curacao or Cyprus or "Czech Republic" or Denmark or Estonia or "Faroe Islands" or Finland or France or "French Polynesia" or Germany or Gibraltar or Greece or Greenland or Guam or "Hong Kong SAR, China" or Hungary or Iceland or Ireland or "Isle of Man" or Israel or Italy or Japan or "Korea, Rep." or Kuwait or Latvia or Liechtenstein or Lithuania or Luxembourg or "Macao SAR, China" or Malta or Monaco or Nauru or Netherlands or "New Caledonia" or "New Zealand" or "Northern Mariana Islands" or Norway or Oman or Panama or Poland or Portugal or "Puerto Rico" or Qatar or Romania or "San Marino" or "Saudi Arabia" or Seychelles or Singapore or "Sint Maarten (Dutch part)" or "Slovak Republic" or Slovenia or Spain or "St. Kitts and Nevis" or "St. Martin (French part)" or Sweden or Switzerland or "Taiwan, China" or "Trinidad and Tobago" or "Turks and Caicos Islands" or "United Arab Emirates" or "United Kingdom" or "United States" or Uruguay or "Virgin Islands (U.S.)").tw,ab. |
| **Limiters** | English language and humans and yr = "2010 -Current" | | | |

However, we will exclude studies involving economic evaluation of reproductive health interventions such as infertility treatment, that are not directly related to perinatal outcomes.

**Comparisons.** The comparator in each economic evaluation study must adhere to the following criteria:

i.  No interventions: Studies comparing the economic outcomes of routine perinatal health care interventions against no interventions.

ii.  Routine interventions: Comparison between routine and new perinatal health interventions/initiatives.

iii. Multiple routine interventions: Studies conducting cost minimisation analyses comparing multiple routine perinatal health interventions.

**Outcomes.** The primary outcomes of our review encompass cost-effectiveness (measured by cost-effectiveness ratio), cost-benefits (return on investment), and/or cost-utility of perinatal health interventions. We will report the absolute costs and effectiveness, benefits or utility gained per intervention and their respective incremental values. Studies solely focusing on the cost of perinatal morbidity and mortality will be excluded.

**Types of studies.** We will incorporate all primary observational studies, including case-control, cohort, cross-sectional designs, and interventional and experimental studies. However, we will exclude reports, qualitative studies, editorials, and conference papers. Additionally, studies not reporting primary research evidence, such as reviews and those studies not pertinent to the economic evaluation aspects of perinatal health interventions, will be excluded.

**Context.** We will restrict our search to high-income countries, as defined by the World Bank, with a gross national income per capita of $13,205 or more in 2023/24 [23].

## Data extraction and analysis

We will use a prior developed Excel spreadsheet to extract data from the included studies. The abstraction process will cover the following aspects:

1. Study characteristics: Author, year of publication, country/setting), study design (prospective cohort, retrospective cohort, case-control, cross sectional and interventional).

2. Study population: Pregnant women, perinatal, neonate, infant, children under five years of age,

3. Participants characteristics: Gestational age at birth (if applicable), birthweight (if applicable), and sample size.,

4. Clinical characteristics: Type of adverse perinatal outcome(s), comorbidities among mothers and or neonates/infants.

5. Outcomes of the study: Cost and effectiveness or benefits (Box 1).

---

### Box 1. Data abstraction sheet

Data elements

1. Author, year
2. Country
3. Study design
4. Study population
5. Sample size
6. Mean/median gestational age at birth (if applicable)
7. Mean/median birthweight (if applicable)
8. Condition/adverse perinatal outcomes

---

9. Intervention(s)/strategy

10. Comparator(s)

11. Economic evaluation type

12. Cost perspective

13. Cost categories/costing items

14. Costing approach/valuation of cost

15. Time horizon (in years)

16. Year of pricing

17. Currency

18. Total cost in local currency

19. Exchange rate to USD in 2022 (using consumer pricing index and purchasing power parities)

20. Overall cost of intervention(s) in USD in 2022

21. Outcomes of interventions (effectiveness or benefits)

22. Model/methods of analysis

23. Incremental cost or Incremental cost effectiveness ratio

24. Average cost/ effectiveness

25. Incremental cost/ effectiveness

We will convert all costs into 2022 United States dollars to facilitate comparison of the incremental cost-effectiveness (ICERs) across the studies. We will employ the Campbell and Cochrane Economics Methods Group (CCEMG) and the Evidence for Policy and Practice Information and Coordinating Centre (EPPI—Centre) cot conversion version 1.6, which utilises the purchasing power parity approach. This approach sources data from the IMF World economic Outlook database [24] and converts all non-US dollar currencies to US dollar currency.

Given the challenges associated with pooling cost and effectiveness/benefits/utility data from different interventions, countries/settings or derived using different methodological approaches, we will provide a narrative and tabular summary of the results. To effectively manage this breadth, we will employ subgroup summarization based on timing across the continuum of maternal and neonatal care: during pregnancy, intrapartum, and the postpartum period, and outcomes such as cost per life-year saved, cost per disability-adjusted life year (DALY) averted, or cost per quality-adjusted life-year (QALY) gained during these three periods (antepartum, intrapartum, and postpartum). Each included outcome will also be presented separately within the respective subgroups.

## Quality of evidence assessment and amendments

We will evaluate the quality of studies using the modified Drummond checklist, a recognised tool for assessing the quality of economic evaluation studies [25]. We will document necessary

protocol amendments in the systematic review's PROSPERO record and publish the updated part of the protocol alongside the full systematic review.

## Discussion

While some studies have examined the effectiveness, benefits, and costs of various perinatal health interventions, there is a lack of systematically summarised evidence. Our review represents the first comprehensive economic evaluation systematically assessing the cost-effectiveness, cost-benefits, and cost-utility of perinatal health interventions from pregnancy to early childhood. In this regard, our review will generate new, comprehensive, and policy-relevant economic evidence to guide health decision-makers in identifying the most effective healthcare interventions for reducing perinatal mortality and morbidity in high-income countries. Furthermore, we firmly believe that our findings will be transferable to other low- and middle-income settings and contribute to achieving SDG target 3.2, which aims to 'end preventable deaths of newborns and children under five years' by 2030 [26].

We anticipate that our review will significantly impact healthcare practices by providing a comprehensive overview of the cost-effectiveness, cost-benefits, and cost-utility associated with perinatal health interventions spanning from pregnancy to the early childhood phase to prevent adverse perinatal outcomes. This valuable information will empower healthcare professionals and potential stakeholders to tailor their strategies based on the most cost-effective or beneficial perinatal health interventions available, inspiring a more effective and efficient healthcare system.

## Supporting information

**S1 Checklist. PRISMA-P (Preferred Reporting Items for Systematic review and Meta-Analysis Protocols) 2015 checklist: Recommended items to address in a systematic review protocol\*.**
(DOC)

## Acknowledgments

We would like to express our sincere gratitude to the Curtin School of Population Health Research and Teaching Incentive scheme for its support. We also thank the reviewers and editor for their insightful comments, which significantly improved the quality of this protocol.

## Author Contributions

**Conceptualization:** Marshall Makate.

**Formal analysis:** Tsegaye G. Haile.

**Funding acquisition:** Marshall Makate.

**Methodology:** Tsegaye G. Haile, Gizachew A. Tessema, Lucas Hertzog, Elizabeth Newnham, Berihun Assefa Dachew, Marshall Makate.

**Supervision:** Gizachew A. Tessema, Lucas Hertzog, Elizabeth Newnham, Berihun Assefa Dachew, Marshall Makate.

**Writing – original draft:** Tsegaye G. Haile.

**Writing – review & editing:** Tsegaye G. Haile, Gizachew A. Tessema, Lucas Hertzog, Elizabeth Newnham, Berihun Assefa Dachew, Marshall Makate.

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
