## [Decision Letter · Decision Letter 0]

8 Mar 2024

PONE-D-23-42050Cost-effectiveness and benefits of perinatal health interventions in high income settings: a protocol for a systematic review of economic evaluationsPLOS ONE

Dear Dr. Haile,

Thank you for submitting your manuscript to PLOS ONE. After careful consideration, we feel that it has merit but does not fully meet PLOS ONE’s publication criteria as it currently stands. Therefore, we invite you to submit a revised version of the manuscript that addresses the points raised during the review process.

Dear Respectable AuthorsThe review is complete. Please send us the revised version as soon as possible. Please respond to all review comments point by point and highlight changes through the text.Cheers

We look forward to receiving your revised manuscript.

Kind regards,

Morteza Arab-Zozani, Ph. D.

Academic Editor

PLOS ONE

Journal Requirements:

Reviewers' comments:

Reviewer's Responses to Questions

**Comments to the Author**

1. Does the manuscript provide a valid rationale for the proposed study, with clearly identified and justified research questions?

Reviewer #1: Partly

Reviewer #2: Partly

Reviewer #3: Yes

2. Is the protocol technically sound and planned in a manner that will lead to a meaningful outcome and allow testing the stated hypotheses?

Reviewer #1: Partly

Reviewer #2: No

Reviewer #3: Yes

3. Is the methodology feasible and described in sufficient detail to allow the work to be replicable?

Reviewer #1: Yes

Reviewer #2: No

Reviewer #3: Yes

4. Have the authors described where all data underlying the findings will be made available when the study is complete?

Reviewer #1: Yes

Reviewer #2: No

Reviewer #3: Yes

5. Is the manuscript presented in an intelligible fashion and written in standard English?

Reviewer #1: No

Reviewer #2: Yes

Reviewer #3: Yes

6. Review Comments to the Author

You may also provide optional suggestions and comments to authors that they might find helpful in planning their study.

Reviewer #1: Publication of protocols and analysis plans is becoming common practice quite rapidly and I feel this is a good development in the light of open science, although the protocol of a review may bear less relevance than the protocol of a planned clinical study. Especially since the protocol was also registered in PROSPERO.

Apart from this, there is quite some room for improvement in the current manuscript, which are summed up below.

- To begin with, I would like to point out the paper series published by a Dutch research group on performing reviews of economic evaluations – papers number 1 and 3 may offer some helpful suggestions: How to prepare a systematic review of economic evaluations for informing evidence-based healthcare decisions: a five-step approach (part 1/3) - https://pubmed.ncbi.nlm.nih.gov/27805469/

- The aim of the review is, reading from the abstract: ‘to comprehensively assess the economic evidence for such interventions in high-income countries’. The authors plan to do this by extracting outcomes such as cost-effectiveness, cost-utility and cost-benefit from the included studies. What is not clear from this paper though is how the authors plan to summarize outcomes. It is to be expected that there will be a large variety of interventions, populations, and outcomes found in the included papers. How do the authors plan to summarize results of a study into the cost-effectiveness of prophylactic aspirin in all pregnant women to prevent complications such as pre-eclampsia, and a study into the cost-benefits of intensified obstetric care to prevent complications during delivery? The authors state that ‘due to the challenges in pooling cost and effectiveness/benefits/utility data from different interventions, countries/settings or derived using different methodological approaches, a narrative and tabular summary of the results will be made’. I agree, pooling will not be possible at all probably, but then the only thing that can be done is presenting enormous tables (because the authors want to include practically everything that has something to do with costs and pregnancy) with no way or structure to draw overall conclusions from these. The idea to perform ‘subgroup analysis based on the type of economic evaluation … for better description of the most cost-effective and beneficial perinatal health interventions’ can probably only help when cost per QALY (or maybe DALY – but not often seen in HIC) are reported, otherwise one cannot compare between interventions/populations etc. Also, it does not make sense to include cost studies as they are not, strictly speaking, ‘economic evidence’ which would enable decision makers to prioritize an intervention. In my opinion, the authors are very ambitious, too ambitious. It would help if there was already a preliminary run of the search terms to see how many hits there are – probably hundreds to thousands?

- From the introduction and discussion sections, it does not become too obvious yet why the researchers chose the high income setting to address the SDG. It is quite well known that the larger part (maybe as much as 90%) of the burden of maternal and child morbidity and mortality lies outside of the high income setting. Although the authors make an argument that also in high income settings there is large differences between population groups, I am not convinced that this systematic review will truly contribute to attaining the SDGs. Would there be a way to extend findings and conclusions in this review to the LMIC setting, that is, when interventions are found to be cost-effective in HIC, may they also be cost-effective in LMIC? Would the review deliver recommendations for LMIC as well? If that were the case, there would be more relevance of this study in relation to the SDG.

- The sub-section on comparisons (p10) puzzles me greatly – I do not understand what is said here.

- English language is a problem – even a non-native speaker as myself spotted a large amount of typos and incorrect sentences. Examples (did not list them all):

o P3 lines 55-56 -> with an estimated 2 million…. occur annually (occurring)

o P3 lines 57 -> SDGs target 3.2 aim to…. (aims to)

o P9 line 152 -> First, two reviewers will first (2x first)

o P9 line 172 -> neonatal mortality, preterm, low birthweight (preterm birth)

o P10 lines 177-178 -> i) no interventions for these routine perinatal health care interventions (I don’t understand this sentence at all)

o P10 lines 189-190 -> we will include …. will be included (phrasing is bad)

o P10 line 196 -> comprise (comprising)

Please run a very thorough check of the language and make sure it reads well.

Reviewer #2: This study reports on “Cost-effectiveness and benefits of perinatal health interventions in high income settings: a protocol for a systematic review of economic evaluations”. The topic is of interest.

- Some suggestions are listed below, which may be useful to the authors as they seek to revise their manuscript.

General comment:

I invite the authors write the paper in the “past tense”.

Abstract

Methodology: Please report the year, the study was conducted, and time framework the studies covered.

Introduction

-This section is written well and sounds good.

Methods

-Like abstract, please report the year, the study was conducted, and time framework the studies covered.

-How was the quality of the studies scored?

-The Table of quality assessment is missing.

Results

-The result section is missing.

Discussion

-This section should rewrite and improve.

-This section is too short.

-The conclusion is missing.

-The limitations and strengths section is missing.

Reviewer #3: The authors have undertaken a significant endeavor to analyze the economic efficiency of health interventions directed at perinatal morbidity and mortality in high-income countries. Such work is indeed praiseworthy and necessary, as it is in line with the Sustainable Development Goals, addressing profound clinical and economic issues. The manuscript's in-depth examination of the disparities in adverse perinatal outcomes across socioeconomic strata is critical for a nuanced understanding of this global issue.

The manuscript notably identifies a significant gap in the literature concerning comprehensive economic evaluations of perinatal health interventions. This point is particularly impactful, with potential to greatly influence policy-making for enhanced health outcomes. The focus on contemporary studies post-2010 is appropriate given the progressive nature of healthcare interventions and economics, and the reviewer supports this approach. Nevertheless, the choice to exclude older studies should be substantiated with a strong rationale, as it could omit historical data essential for trend analysis. The authors are thus encouraged to elaborate on this choice and suggest ways to balance the need for current data with the value of historical trends.

While the manuscript adeptly handles high-income country scenarios, its findings' applicability to low- and middle-income countries (LMICs) is not discussed. Given the disproportionate burden of perinatal morbidity and mortality in these settings, the authors should consider the transferability of effective interventions from high-income contexts to LMICs, taking into account the differences in resources and healthcare systems.

The manuscript would gain depth by addressing the possible limitations inherent in its current scope, especially concerning the exclusion of pre-2010 data. Including a discussion about the trends in healthcare costs, the effects of historical health policies, and the longevity of intervention outcomes would greatly contribute to understanding the current economic milieu of perinatal health interventions.

In conclusion, the manuscript provides a comprehensive and well-organized study protocol that adeptly addresses a significant health issue, commendably aligning with the objectives outlined in the Sustainable Development Goals, particularly those targeting profound clinical and economic challenges. While the manuscript currently does not include the results of the study, incorporating such findings in the future would significantly enhance the value and richness of the paper. Nonetheless, as it stands, the protocol offers a robust framework for the investigation of the economic efficiency of interventions aimed at improving perinatal health outcomes in high-income countries. It is advisable for the authors to expand their discussion to incorporate the issues that have been highlighted. Delving deeper into these areas would not only broaden the manuscript's scope and deepen its analytical insights but also elevate its significance and contribution to the wider academic and policy-making discourse.

7. PLOS authors have the option to publish the peer review history of their article (what does this mean?). If published, this will include your full peer review and any attached files.

Reviewer #1: No

Reviewer #2: No

Reviewer #3: No

---

## [Author Response · Author response to Decision Letter 0]

14 Apr 2024

10 April 2024

Re: Cost-effectiveness and benefits of perinatal health interventions in high income settings: a protocol for a systematic review of economic evaluations (Manuscript #: PONE-D-23-42050)

Prof Morteza Arab-Zozani

Academic Editor

PLOS ONE

Re: Response to reviewers’ comments

Dear Morteza,

We would like to express our gratitude for the opportunity to submit our revised protocol entitled Cost-effectiveness and benefits of perinatal health interventions in high income settings: a protocol for a systematic review of economic evaluations (manuscript ID: PONE-D-23-42050). We also thank the reviewers for their constructive comments and the time and effort they invested in evaluating our work. 

We have addressed the comments of the reviewers’ and the editor, responded to them point-by- point, revised the manuscript accordingly, and presented the revisions in track changes. We hope that you will find the revised protocol acceptable for publication.

Thank you for your time and considerations.

Kind regards,

Tsegaye Gebremedhin Haile (MSc, MPH, PhD Candidate), corresponding author

Dr Gizachew A Tessema, Dr Lucas Hertzog, Dr Elizabeth Newnham, Dr Berihun Dachew, Dr Marshall Makate

 

Response to reviewers’ comments 

Reviewer 1

Publication of protocols and analysis plans is becoming common practice quite rapidly and I feel this is a good development in the light of open science, although the protocol of a review may bear less relevance than the protocol of a planned clinical study. Especially since the protocol was also registered in PROSPERO.

Response: Dear reviewer, thank you so much for your time, comments, and feedback on the improvement of our protocol, which is crucial for our current review. We believe this protocol holds paramount importance for our ongoing review, given its potential to significantly enhance the quality of evidence generation, synthesis, and reporting. Additionally, its publication will ensure transparency and reproducibility, fostering collaboration and facilitating the dissemination of best practices within the scientific community, ultimately advancing the field. When it is published alongside its corresponding review, we strongly believe it will serve as the premier guide for those seeking to conduct systematic reviews in the field of economic evaluations and related areas.

1. To begin with, I would like to point out the paper series published by a Dutch research group on performing reviews of economic evaluations – papers number 1 and 3 may offer some helpful suggestions: How to prepare a systematic review of economic evaluations for informing evidence-based healthcare decisions: a five-step approach (part 1/3) - https://pubmed.ncbi.nlm.nih.gov/27805469/

Response: Thank you for bringing this resource to our attention. We have reviewed it thoroughly and found it to be valuable for both our ongoing review and our future endeavours. 

2. The aim of the review is, reading from the abstract: ‘to comprehensively assess the economic evidence for such interventions in high-income countries’. The authors plan to do this by extracting outcomes such as cost-effectiveness, cost-utility and cost-benefit from the included studies. What is not clear from this paper though is how the authors plan to summarize outcomes. It is to be expected that there will be a large variety of interventions, populations, and outcomes found in the included papers. How do the authors plan to summarize results of a study into the cost-effectiveness of prophylactic aspirin in all pregnant women to prevent complications such as pre-eclampsia, and a study into the cost-benefits of intensified obstetric care to prevent complications during delivery? The authors state that ‘due to the challenges in pooling cost and effectiveness/benefits/utility data from different interventions, countries/settings or derived using different methodological approaches, a narrative and tabular summary of the results will be made’. I agree, pooling will not be possible at all probably, but then the only thing that can be done is presenting enormous tables (because the authors want to include practically everything that has something to do with costs and pregnancy) with no way or structure to draw overall conclusions from these. The idea to perform ‘subgroup analysis based on the type of economic evaluation … for better description of the most cost-effective and beneficial perinatal health interventions’ can probably only help when cost per QALY (or maybe DALY – but not often seen in HIC) are reported, otherwise one cannot compare between interventions/populations etc. Also, it does not make sense to include cost studies as they are not, strictly speaking, ‘economic evidence’ which would enable decision makers to prioritize an intervention. In my opinion, the authors are very ambitious, too ambitious. It would help if there was already a preliminary run of the search terms to see how many hits there are – probably hundreds to thousands?

Response: Thank you for your critical comments and insightful suggestions. We fully acknowledge that our review encompasses a diverse range of interventions, populations, and outcomes. To effectively manage this breadth, we will employ subgroup summarization based on timing across the continuum of maternal and neonatal care: during pregnancy, intrapartum, and the postpartum period, and outcomes such as cost per life-year saved, cost per disability-adjusted life year (DALY) averted, or cost per quality-adjusted life-year (QALY) gained during these three periods (antepartum, intrapartum, and postpartum). Each included outcome will also be presented separately within the respective subgroups. This change has been found on page 11, lines 229 to 235.

We will also exclude partial economic evaluations and we revised the inclusion and exclusion criteria accordingly . 

After conducting a preliminary run, we have identified approximately 5,180 studies after removing duplicates for title and abstract screening. We are confident in our ability to manage this volume of studies effectively. We believe that our review is comprehensive and systematic, and we are committed to generating valuable evidence for decision-makers, researchers, clinicians, and other stakeholders in the field. Our aim is to provide a comprehensive understanding of the topic and support informed decision-making processes. Therefore, we do not consider our approach to be overly ambitious.

3. From the introduction and discussion sections, it does not become too obvious yet why the researchers chose the high income setting to address the SDG. It is quite well known that the larger part (maybe as much as 90%) of the burden of maternal and child morbidity and mortality lies outside of the high income setting. Although the authors make an argument that also in high income settings there is large differences between population groups, I am not convinced that this systematic review will truly contribute to attaining the SDGs. Would there be a way to extend findings and conclusions in this review to the LMIC setting, that is, when interventions are found to be cost-effective in HIC, may they also be cost-effective in LMIC? Would the review deliver recommendations for LMIC as well? If that were the case, there would be more relevance of this study in relation to the SDG.

Response: Thank you for your insights. We agree that a significant portion of maternal, neonatal, and child morbidity and mortality occurs in low- and middle-income countries. Consequently, extensive research has been conducted in these regions. However, due to the assumption that high-income countries have already addressed the SDG targets, less attention has been given to them, leading to large disparities among population groups. Therefore, examining the evidence in these high-income regions is of paramount importance. Moreover, findings from interventions that are effective in high-income countries can also be transferable to low- and middle-income countries, presented as follows: 

“While high-income countries have made substantial strides in perinatal health, disparities persist among various groups. Addressing these disparities necessitates tailored interventions that consider specific challenges. Unfortunately, the implementation of cost-effective interventions aimed at mitigating these disparities has not received sufficient attention within the healthcare system. Recognizing this oversight early on is vital for future cost savings in the health sector. Moreover, differences in healthcare infrastructure, resources, access to care, and policy priorities contribute significantly to perinatal health outcome disparities between high-income and other settings. Therefore, it is crucial to examine evidence on the cost-effectiveness and benefits of perinatal health interventions separately. 

In today's context, with perinatal morbidity and mortality remaining significant in low- and middle-income countries, the majority of perinatal health intervention studies focus on these regions, resulting in a lack of comprehensive evidence on the economic evaluations of perinatal health in high-income nations.” The change can be found on page 4, lines 92 to 104.

4. The sub-section on comparisons (p10) puzzles me greatly – I do not understand what is said here.

Response: We sincerely apologise for any confusion regarding the presentation of the comparison group. We have revised the comparison section of the PICO approach accordingly. The comparators will include no intervention, routine interventions, multiple interventions, or no comparison at all depending on the intervention assessed. Below are detailed descriptions of these comparisons:

“The comparator in each economic evaluation study must adhere to the following criteria: 

i) No intervention: Studies comparing the economic outcomes of routine perinatal health care interventions against no interventions. 

ii) Routine interventions: Comparison between routine interventions and new perinatal health interventions/initiatives. 

iii) Multiple routine interventions: Studies conducting cost minimisation analyses comparing multiple routine perinatal health interventions. 

iv) No comparator: Economic evaluations focusing solely on the costs of perinatal health interventions without specific comparators.”

We believe our revisions are clear for the readers and adequately address the needs of reviewer. The changes can be found on page 10, lines 183 to 191.

5. English language is a problem – even a non-native speaker as myself spotted a large amount of typos and incorrect sentences. Examples (did not list them all):

• P3 lines 55-56 -> with an estimated 2 million…. occur annually (occurring)

• P3 lines 57 -> SDGs target 3.2 aim to…. (aims to)

• P9 line 152 -> First, two reviewers will first (2x first)

• P9 line 172 -> neonatal mortality, preterm, low birthweight (preterm birth)

• P10 lines 177-178 -> i) no interventions for these routine perinatal health care interventions (I don’t understand this sentence at all)

• P10 lines 189-190 -> we will include …. will be included (phrasing is bad)

• P10 line 196 -> comprise (comprising)

Please run a very thorough check of the language and make sure it reads well.

Response: Thank you for your great attention and pointing out this. We have carefully reviewed all sections of the document, addressing grammar errors, typos, and rephrasing paragraphs, sentences, and punctuation where necessary. These revisions have been implemented throughout the entire protocol.

Reviewer #2: This study reports on “Cost-effectiveness and benefits of perinatal health interventions in high income settings: a protocol for a systematic review of economic evaluations”. The topic is of interest.

• Some suggestions are listed below, which may be useful to the authors as they seek to revise their manuscript.

1. General comment:

• I invite the authors write the paper in the “past tense”.

Response: Dear reviewer, thank you very much for dedicating your time to review our protocol and for providing your valuable comments, suggestions, and feedback. Your input will significantly enhance both our protocol and the forthcoming review. However, it’s important to note that this document outlines our protocol for conducting the actual review in the near future, following the incorporation of comments, suggestions, and the expertise of reviewers, as well as the publications of the protocol. Consequently, we present the methods in the future tense. During the final manuscript’s preparation and presentation of results, we will appropriately revise the tense to reflect past actions. Therefore, we kindly request you evaluate this document as a protocol and not as the actual review or its results.

2. Abstract

• Methodology: Please report the year, the study was conducted, and time framework the studies covered.

Response: Thank you for your comments. Our review will cover studies published from the first of January 2010 to the date of search, which will be reported in detail during the final result presentations. 

3. Introduction

• This section is written well and sounds good.

Response: Thank you for your observations.

4. Methods

• Like abstract, please report the year, the study was conducted, and time framework the studies covered.

• How was the quality of the studies scored?

• The Table of quality assessment is missing.

Response: Thank you for your comments. Our review will encompass studies published from January 1, 2010, up to the date of the search. We will evaluate the quality of studies using the modified Drummond checklist, which is a recognised tool for assessing the quality of economic evaluation studies(Drummond, Sculpher et al. 2015).

5. Results

• The result section is missing.

Response: Thank you for your observations. However, please note that this document outlines our protocol, and the results have not yet been presented.

6. Discussion

• This section should rewrite and improve.

• This section is too short.

• The conclusion is missing.

• The limitations and strengths section is missing.

Response: As mentioned previously, we acknowledge that the discussion section may currently appear somewhat shallow. However, it's important to emphasise that the results of our review have not yet been presented. We anticipate that the depth and breadth of the discussion will be substantially strengthened during the result presentation phase, based on the findings we will obtain. Furthermore, our final manuscript will incorporate a concise conclusion, along with an evaluation of both the limitations and strengths of our review.

Reviewer #3: The authors have undertaken a significant endeavor to analyze the economic efficiency of health interventions directed at perinatal morbidity and mortality in high-income countries. Such work is indeed praiseworthy and necessary, as it is in line with the Sustainable Development Goals, addressing profound clinical and economic issues. The manuscript's in-depth examination of the disparities in adverse perinatal outcomes across socioeconomic strata is critical for a nuanced understanding of this global issue.

1. The manuscript notably identifies a significant gap in the literature concerning comprehensive economic evaluations of perinatal health interventions. This point is particularly impactful, with potential to greatly influence policy-making for enhanced health outcomes. The focus on contemporary studies post-2010 is appropriate given the progressive nature of healthcare interventions and economics, and the reviewer supports this approach. Nevertheless, the choice to exclude older studies should be substantiated with a strong rationale, as it could omit historical data essential for trend analysis. The authors are thus encouraged to elaborate on this choice and suggest ways to balance the need for current data with the value of historical trends.

Response: Dear reviewer, thank you so much for your insightful comments, suggestions, and feedback. We fully recognize the importance of data predating 2010 for conducting trend analysis. However, our review will synthesise the recent evidence on the cost-effectiveness, cost-utility, and cost-benefit of perinatal health interventions in high-income settings. 

The healthc

---

## [Decision Letter · Decision Letter 1]

17 May 2024

PONE-D-23-42050R1Cost-effectiveness and benefits of perinatal health interventions in high income settings: a protocol for a systematic review of economic evaluationsPLOS ONE

Dear Dr. Haile,

Thank you for submitting your manuscript to PLOS ONE. After careful consideration, we feel that it has merit but does not fully meet PLOS ONE’s publication criteria as it currently stands. Therefore, we invite you to submit a revised version of the manuscript that addresses the points raised during the review process.

We look forward to receiving your revised manuscript.

Kind regards,

Miquel Vall-llosera Camps

Senior Staff Editor

PLOS ONE

Journal Requirements:

Reviewers' comments:

Reviewer's Responses to Questions

**Comments to the Author**

1. Does the manuscript provide a valid rationale for the proposed study, with clearly identified and justified research questions?

Reviewer #1: Yes

Reviewer #3: Yes

2. Is the protocol technically sound and planned in a manner that will lead to a meaningful outcome and allow testing the stated hypotheses?

Reviewer #1: Yes

Reviewer #3: Yes

3. Is the methodology feasible and described in sufficient detail to allow the work to be replicable?

Reviewer #1: Yes

Reviewer #3: Yes

4. Have the authors described where all data underlying the findings will be made available when the study is complete?

Reviewer #1: Yes

Reviewer #3: Yes

5. Is the manuscript presented in an intelligible fashion and written in standard English?

Reviewer #1: Yes

Reviewer #3: Yes

6. Review Comments to the Author

You may also provide optional suggestions and comments to authors that they might find helpful in planning their study.

**Reviewer #1:** The authors have made a substantial effort to improve their manuscript and have taken the suggestions provided by the reviewers to heart. I have one comment still with regard to the definition of the comparators give the updated scope of the review: the authors have chosen to not include partial economic evaluations in their planned review anymore, which I feel is a sensible decision. However, since a full economic evaluation would always include a comparative element (see also the clear definitions in this paper: https://www.ncbi.nlm.nih.gov/pmc/articles/PMC8424074/ ) - the fourth comparator on p10 of the clean manuscript, stating 'No comparators: Economic evaluations focusing solely on the costs of perinatal health interventions without specific comparators.' - is not a valid comparison here - this could only be a relevant thing to study when partial economic evaluations would be included. So this 4th option should be removed from the list.

Last remark: given that scope and search terms have been updated, will this also be aligned in the prospero protocol, that is, will prospero be updated accordingly?

**Reviewer #3:** This review is expected to notably affect healthcare practices by summarizing the financial impacts and benefits of perinatal interventions, aiding in the optimization of strategies to avert negative perinatal outcomes.

7. PLOS authors have the option to publish the peer review history of their article (what does this mean?). If published, this will include your full peer review and any attached files.

Reviewer #1: **Yes: **Antoinette D.I. van Asselt

Reviewer #3: **Yes: **Abu Ahmed

---

## [Author Response · Author response to Decision Letter 1]

21 May 2024

21st May 2024

Re: Cost-effectiveness and benefits of perinatal health interventions in high-income settings: a protocol for a systematic review of economic evaluations (Manuscript #: PONE-D-23-42050R1)

Prof Miquel Vall-llosera Camps

Senior Staff Editor, PLOS ONE

Re: Response to reviewers’ comments

Dear Professor Miquel,

We are pleased to submit our revised version protocol entitled Cost-effectiveness and benefits of perinatal health interventions in high-income settings: a protocol for a systematic review of economic evaluations (manuscript ID: PONE-D-23-42050R1). We greatly appreciate the reviewers' comments and feedback, which will enhance our upcoming review.

We have addressed all the comments, incorporated the suggestions, responded to them point-by-point, revised the protocol accordingly, and presented the revisions in track changes. At this stage, we hope that you find the revised protocol acceptable for publication.

 

Response to reviewers’ comments 

Reviewer 1

The authors have made a substantial effort to improve their manuscript and have taken the suggestions provided by the reviewers to heart. 

Q1: I have one comment still with regard to the definition of the comparators give the updated scope of the review: the authors have chosen to not include partial economic evaluations in their planned review anymore, which I feel is a sensible decision. However, since a full economic evaluation would always include a comparative element (see also the clear definitions in this paper: https://www.ncbi.nlm.nih.gov/pmc/articles/PMC8424074/ ) - the fourth comparator on p10 of the clean manuscript, stating 'No comparators: Economic evaluations focusing solely on the costs of perinatal health interventions without specific comparators.' - is not a valid comparison here - this could only be a relevant thing to study when partial economic evaluations would be included. So this 4th option should be removed from the list.

Response: Dear reviewer, thank you so much for reviewing our second version protocol. Your comments and feedback greatly improve our protocol and for our planned actual review. We have removed the fourth comparator 'No comparators: Economic evaluations focusing solely on the costs of perinatal health interventions without specific comparators' as we will include the full economic evaluations. We have also updated the search terms to focus solely on the full economic evaluations. The change has been found on page pages 8 and 10.

Q2. Last remark: given that scope and search terms have been updated, will this also be aligned in the prospero protocol, that is, will prospero be updated accordingly?

Response: Thanks for the remarks. We have mentioned the updates on the PROSPERO registrations (PROSPERO registration # CRD42023432232).

Reviewer 3

Recommendations: This review is expected to notably affect healthcare practices by summarizing the financial impacts and benefits of perinatal interventions, aiding in the optimization of strategies to avert negative perinatal outcomes.

Reply: Dear reviewer, thank you very much for highlighting the importance of our upcoming review.

---

## [Decision Letter · Decision Letter 2]

19 Jun 2024

Cost-effectiveness and benefits of perinatal health interventions in high-income settings: a protocol for a systematic review of economic evaluations

PONE-D-23-42050R2

Dear Dr. Haile,

We’re pleased to inform you that your manuscript has been judged scientifically suitable for publication and will be formally accepted for publication once it meets all outstanding technical requirements.

Kind regards,

Dr. Cristóbal Cuadrado N.

Academic Editor

PLOS ONE

Additional Editor Comments (optional):

Thank you for adequately addressing the commentaries and suggestions of the reviewers and editors. The manuscript is now ready to be accepted for publication.

Reviewers' comments:

Reviewer's Responses to Questions

**Comments to the Author**

1. Does the manuscript provide a valid rationale for the proposed study, with clearly identified and justified research questions?

Reviewer #1: Yes

2. Is the protocol technically sound and planned in a manner that will lead to a meaningful outcome and allow testing the stated hypotheses?

Reviewer #1: Yes

3. Is the methodology feasible and described in sufficient detail to allow the work to be replicable?

Reviewer #1: Yes

4. Have the authors described where all data underlying the findings will be made available when the study is complete?

Reviewer #1: Yes

5. Is the manuscript presented in an intelligible fashion and written in standard English?

Reviewer #1: Yes

6. Review Comments to the Author

You may also provide optional suggestions and comments to authors that they might find helpful in planning their study.

Reviewer #1: I would like to thank the authors for their respectful responses, I have no further comments on the revised version

7. PLOS authors have the option to publish the peer review history of their article (what does this mean?). If published, this will include your full peer review and any attached files.

Reviewer #1: No

---

## [Editor Report · Acceptance letter]

24 Jun 2024

PONE-D-23-42050R2 

PLOS ONE

Dear Dr. Haile, 

I'm pleased to inform you that your manuscript has been deemed suitable for publication in PLOS ONE. Congratulations! Your manuscript is now being handed over to our production team.

Kind regards, 

on behalf of

Dr. Cristóbal Cuadrado 

Academic Editor

PLOS ONE